# A Novel DLLME Method Involving a Solidifying Hydrophobic Deep Eutectic Solvent Using Hydrophilic Deep Eutectic Solvent as the Dispersant for the Determination of Polychlorinated Biphenyls in Water Samples

**DOI:** 10.3390/molecules29153480

**Published:** 2024-07-25

**Authors:** Chunlong Peng, Shuochen Zhang, Xin Li

**Affiliations:** 1Institute of Natural Medicine and Health Products, School of Pharmaceutical Sciences, Taizhou 318000, China; pengcl@tzc.edu.cn (C.P.); zhangsc0317@163.com (S.Z.); 2Zhejiang Provincial Key Laboratory of Plant Evolutionary Ecology and Conservation, Taizhou University, Taizhou 318000, China; 3School of Food and Engineering, Shaanxi University of Science and Technology, Xi’an 710021, China

**Keywords:** DLLME, hydrophobic deep eutectic solvent, hydrophilic deep eutectic solvent, polychlorinated biphenyls, water samples

## Abstract

This paper presents a novel dispersive liquid–liquid microextraction (DLLME) method that employs solidified hydrophobic deep eutectic solvent (DES) with hydrophilic DES acting as the dispersant. The aim is to enrich polychlorinated biphenyls (PCBs) from water samples for subsequent determination by gas chromatography–mass spectrometry. The effects of both the hydrophobic DES as the extractant and the hydrophilic DES as the dispersant were thoroughly investigated. Optimization of the key factors influencing extraction efficiency was performed, and the method was subsequently validated. Specifically, a hydrophobic DES called DES2, prepared by combining thymol and decanoic acid in a molar ratio of 3:2, was selected as the extraction solvent. Meanwhile, a hydrophilic DES named DES6, prepared from choline chloride and acetic acid in a molar ratio of 1:2, was chosen as a dispersant. Under the optimal extraction conditions, the developed method exhibited excellent linearity over the concentration range of 0.01–5.0 µg/L, low limits of detection ranging from 3.0 to 5.1 ng/L, relative standard deviations less than 4.1%, and enrichment factors between 182 and 204 for PCBs. Finally, the effectiveness of the developed method was successfully demonstrated through residue determination of PCBs in water samples.

## 1. Introduction

Polychlorinated biphenyls (PCBs) are a group of human-made chlorinated chemical compounds that were extensively used in various industrial sectors owing to their exceptional physicochemical properties [1]. However, due to their inertness and potential threats to public health, PCBs are classed as persistent organic pollutants according to the Stockholm Convention [2,3]. PCBs have the ability to readily migrate through different environmental media, leading to numerous ecotoxicological concerns [2,4,5]. Among these concerns, water pollution caused by PCBs is particularly alarming [5,6,7]. However, accurately assessing the extent of such pollution is challenging due to the fact that PCB concentrations in water are normally found at very low levels, attributable to their strong hydrophobicity. Consequently, prior to the analysis of PCBs in sample solutions, sample preparation is essential to enhance detection sensitivity.

Sample preparation is an important step in most analytical procedures, as it helps to minimize matrix effects and concentrate the target analytes [8,9,10]. Dispersive liquid–liquid microextraction (DLLME), originally developed by Rezaee et al., is a technique based on the traditional liquid–liquid extraction method [11]. In DLLME, a mixture of extractant and dispersant is transferred into the aqueous sample. The extractant quickly disperses into small droplets in the aqueous phase, greatly enhancing the extraction efficiency. Due to its easy operation, high sensitivity, and reduced consumption of organic solvents, DLLME has gained popularity as a sample preparation method, and it is often employed for isolating and concentrating analytes from water matrices [10,12,13].

DLLME based on solidification of floating organic droplets (DLLME-SFOD) is a more recent variant of DLLME; by this method, the floating solvent used for extraction can be easily collected from the aqueous phase after solidifying at low temperatures [14]. This innovation enhances the convenience and efficiency of the extraction process.

In DLLME-SFOD, the selection of a suitable extractant is crucial for achieving high extraction efficiency. The extractant must possess specific properties such as immiscibility with water, a density lower than water, a melting point near room temperature, and a high partition coefficient for the analytes [15,16]. Deep eutectic solvents (DESs) have emerged as a promising group of solvents that meet these requirements. DESs are synthesized by combining a hydrogen bond donor (HBD) with a hydrogen bond acceptor (HBA), and the components of DESs are held together by intermolecular hydrogen bonds. DESs can be customized for different analytes by carefully choosing the appropriate HBD and HBA, as well as their molar ratio [15,17]. Most DESs exhibit a liquid state at room temperature and possess properties similar to ionic liquids. Moreover, some DESs are cost-effective, have low toxicity, and are biodegradable, making them attractive alternatives to traditional organic solvents in microextraction [18,19,20].

The application of DESs in DLLME was firstly reported by Farajzadeh et al. in 2016 for analysis of pesticides in foods [21,22]. In this report, a hydrophobic DES used as the extractant was prepared by mixing choline chloride (ChCl) and chlorophenol. After that, the number of reports employing hydrophobic DESs in DLLME has increased rapidly. Moreover, various techniques were proposed to facilitate the dispersion of the DESs, including vortex-assisted and ultrasound-assisted diffusion [23,24]. In recent years, DESs applied in DLLME-SFOD as extractants are found to be applicable, and certain hydrophobic DESs have been successfully employed as extractants in DLLME-SFOD for the extraction of different analytes, such as metals, lignin, and environmental pollutants [25,26,27,28,29].

In DLLME, a dispersant is typically employed to aid in the dispersion of the extractant into small droplets. Commonly used dispersants include methanol, acetonitrile, and acetone. However, there is growing interest among analytical researchers in exploring greener alternatives to these organic dispersants, and deep eutectic solvents (DESs) have emerged as attractive options. Several hydrophilic DESs have been reported to be effective as dispersants in DLLME [30,31,32]. This means that certain DESs can serve as either the extractant or the dispersant in DLLME, depending on their specific properties.

Indeed, while the DLLME method based on the solidification of hydrophobic DESs using hydrophilic DESs as dispersants (DLLME-SHDES-HDESD) shows promise, its reported applications are still relatively limited. Further research is necessary to fully explore and understand the potential of this approach. This includes investigating its applicability to different analytes and matrices, optimizing extraction conditions, and assessing its performance compared to other extraction techniques. The development of new hydrophobic and hydrophilic DESs with enhanced extraction properties could also contribute to expanding the scope of this method. Overall, more studies are needed to fully exploit the potential of DLLME with solidified hydrophobic DESs and hydrophilic DESs as a versatile and green sample preparation technique.

This study aims to propose a novel DLLME-SHDES-HDESD method followed by gas chromatography–mass spectrometry (GC-MS) for the analysis of PCBs in water samples. The main focus is to investigate the key experimental factors that affect extraction efficiency. Following optimization of the extraction conditions, the developed method is validated and subsequently applied for determination of PCBs in real water samples.

## 2. Results and Discussion

### 2.1. Characterization of DESs

It is known that hydrogen bonding between the HBD and HBA is the primary force for the formation of DESs. The existence of hydrogen bonding in three hydrophilic DESs (DES5, DES6, and DES7) has been reported in our previous study [33]. In this study, to elucidate the existence of hydrogen-bonding interactions in the hydrophobic DESs, FT-IR and ^1^H-NMR experiments of the representative DES2 were conducted. To begin, FT-IR spectra of DES2, thymol, and decanoic acid were recorded. The O-H stretching vibration peak of DES2 was observed to shift to a higher wavenumber (3398.98 cm^−1^) compared to thymol (3172.18 cm^−1^), as shown in Appendix A. This change might have been due to a subtle change in the force constant that was caused by the decreased electron cloud density [34,35]. This finding indicates formation of hydrogen bonds between thymol and decanoic acid in DES2.

Additionally, the chemical structure of DES2 was further investigated using ^1^H-NMR. DES2 was prepared in deuterated dimethyl sulfoxide (DMSO-d_6_) before analysis. As shown in the spectra presented in Appendix A, all peaks observed in DES2 can be assigned to thymol and decanoic acid, without any additional peaks. This indicates that no chemical reactions occurred during the formation of DES2. The absence of any new peaks suggests that hydrogen bonding was formed between the hydroxyl group of thymol and the carboxyl group of decanoic acid.

The physicochemical properties of four hydrophobic (DES1, DES2, DES3, and DES4) and three hydrophilic (DES5, DES6 and DES7) DESs, including their melting points and densities, are summarized in Table 1. The data indicate that all hydrophobic DESs have a density lower than water and a melting point near room temperature. Moreover, the solid–liquid diagram for a mixture of thymol and decanoic acid revealed that each hydrophobic DES has a lower melting point compared with its pure components (Appendix A). Consequently, these hydrophobic DESs are suitable candidates for use as extraction solvents in DLLME-SFOD.

### 2.2. Optimization of the DLLME-SHDES-HDESD Procedure

To optimize the main experimental factors, a DLLME procedure was utilized. During the process, the extraction efficiency was evaluated by comparison of the peak area of the PCBs. However, it is important to note that a modification was made to the conventional DLLME method. Specifically, when a hydrophilic DES was used as the dispersant, the water sample was transferred into the centrifuge tube containing a mixture of extractant and dispersant solvents.

#### 2.2.1. Effect of Extractant Type and Extractant Volume

Four hydrophobic DESs (DES1, DES2, DES3, and DES4) were initially tested as candidate extractants. The results of the experiments indicated that DES2 provided the highest extraction efficiency compared to the other DESs (Figure 1a). Therefore, DES2 was selected as the extractant for further experiments.

Additionally, the effect of extractant volume on extraction efficiency was evaluated by conducting experiments using varying volumes of DES2 ranging from 50 to 125 μL. The results showed that a larger peak area of the PCBs was obtained when 50 μL of DES2 was used (Figure 1b). This observation can be attributed to the dilution effect. Hence, a smaller extractant volume is usually preferable to increase analyte enrichment [15]. Based on these findings, 50 μL of DES2 was chosen for use in subsequent experiments.

#### 2.2.2. Effect of Dispersant Type and Dispersant Volume

The selection of an appropriate dispersant is essential for carrying out an efficient DLLME procedure. The dispersant must possess excellent miscibility with the extraction solvent and aqueous solution. In this experiment, various dispersants, including methanol (MeOH), acetonitrile (ACN), acetone, and three hydrophilic DESs (DES5, DES6, and DES7), were investigated. The results presented in Figure 2a indicate that DES6 yielded a higher extraction efficiency compared to the other dispersants. Furthermore, during the experimental process, the occurrence of hydrophilic DES decomposition was observed following addition of the water sample. This decomposition of the hydrophilic DES could facilitate dispersion of the extractant, leading to enhanced extraction efficiency [32]. Based on these findings, DES6 was selected as the preferred dispersant for further experiments.

Dispersant volume is another factor that affects extraction efficiency in DLLME. In this study, the effect of different volumes of DES6 (150, 300, 450 and 600 µL) on extraction efficiency was evaluated. Figure 2b shows that, as DES6 volume increased from 150 to 300 μL, the extraction efficiency also increased. This can be attributed to the improved dispersing effect resulting from the higher dispersant content. However, due to co-solvency, when the DES6 volume increases from 300 to 600 μL, the peak area of the targeted PCBs decreases. Thus, 300 μL of DES6 was chosen for use in subsequent experiments.

#### 2.2.3. Effect of pH and Salt Addition

The pH of the sample solution can have an impact on the charged state of analytes and the distribution coefficient of the analytes between two phases, which in turn affects their solubility in the extraction solvent [36,37]. In this study, the extraction efficiency of the PCBs was studied by adjusting the pH of the water sample to different values (3, 5, 7, and 9) using solutions of sodium hydroxide or sulfuric acid. The results presented in Figure 3a indicate that the extraction efficiency of the PCBs remains relatively constant across the pH range of 3 to 9. This suggests that the solubility of the targeted PCBs in the extractant is not significantly affected by changes in pH within this range. Hence, it was determined that there was no need to adjust the pH of the water samples, since the pH values of all the samples studied fell in the range of 6 to 8.

Adding salt into aqueous samples before extraction can affect the extraction efficiency of analytes due to the salting-out or salting-in effect [38,39]. In this study, the effect of salt addition on extraction efficiency was evaluated by adding different concentrations of sodium chloride (0%, 5%, 10%, and 15%, *w*/*v*). The results in Figure 3b indicate that there is no obvious change in the extraction efficiency when the sodium chloride concentration is in the range of 0–5%. However, a slight decrease in extraction efficiency is observed as the sodium chloride concentration increases from 5% to 15%. This decrease in extraction efficiency at higher salt concentrations might be attributed to the salt-out effect [40]. Therefore, it was determined that salt addition was unnecessary in the developed method.

#### 2.2.4. Effect of Ultrasounication Time

Ultrasonic radiation is known to accelerate the mass-transfer process of analytes to the liquid phase through the application of alternating acoustic pressure [41]. This can enhance extraction efficiency and reduce extraction time. The effect of ultrasonication time ranging from 1 to 5 min was studied, and the data presented in Figure 4 indicate that a 2 min ultrasonication time was sufficient to complete the extraction procedure. Thus, a 2 min ultrasonication time was selected for use in the following experiments.

### 2.3. Method Validation

The practicality of the developed method was assessed by applying it to the determination of PCBs in a series of spiked water samples. Under the optimized conditions, several performance parameters were evaluated, including linearity range (LR), determination coefficient (R^2^), limit of detection (LOD), limit of quantitation (LOQ), precision, and enrichment factors (EFs). The results are tabulated in Table 2.

The developed method demonstrated good linearity (R^2^ ≥ 0.9985) over the concentration range of 0.01 to 5.0 μg/L for the seven targeted PCBs. The EFs ranged from 182 to 204, indicating efficient extraction and enrichment of the analytes. The LODs were calculated based on a signal-to-noise (S/N) ratio of 3 and ranged from 3.0 to 5.1 ng/L. The LOQs, determined at an S/N ratio of 10, ranged from 10.1 to 17.0 ng/L. These values indicate the sensitivity of the method for detecting and quantifying PCBs. Furthermore, a repeatability study was conducted using samples spiked with each targeted PCB at 0.1 μg/L (*n* = 5). The obtained relative standard deviations (RSDs) were below 4.1%, demonstrating good precision of the developed method.

### 2.4. Comparison of DLLME-SHDES-HDESD with Other Microextraction Methods

The characteristics of the DLLME-SHDES-HDESD method were compared with other previously reported microextraction methods for the preconcentration of PCBs. The results of this comparison are tabulated in Table 3. The LOD achieved by the developed method was found to be comparable to or lower than those of other extraction methods [42,43,44,45,46]. The percentage RSD (%RSD) of the developed method was comparable to that of the LDS-USAEME and SSME methods but lower than that of other methods [42,43,44,45]. In terms of enrichment factor, the developed method exhibited a high enrichment factor, indicating efficient extraction and concentration of PCBs from water samples. Furthermore, the extractant and dispersant used in the developed method were reported to be more environmentally friendly compared to the solvents employed in other methods [42,43,44,45,46]. Taken together, the developed method is characterized as simple, sensitive, and green, making it a promising option for the preconcentration of PCBs in water samples.

### 2.5. Real Water Sample Analysis

To evaluate the matrix effects of real water samples on the extraction efficiency, the developed method was employed to determine PCBs in tap and river water, as well as industrial wastewater samples. Under optimized conditions, the percentage recoveries of PCBs spiked into real water samples at concentrations of 0.1 and 1 µg/L were evaluated. The data, as shown in Table 4, revealed that the spiked recoveries for three water samples ranged from 92.58% to 103.9%, with RSDs between 2.1% and 4.5%. The results indicate satisfactory recovery ranges for PCBs in real water samples. Moreover, the recovery values for these three water samples were almost identical, indicating that the method’s performance was not obviously affected by sample matrices.

## 3. Materials and Methods

### 3.1. Materials and Reagents

In this study, the focus is on monitoring seven indicator PCBs, specifically congener numbers 28, 52, 101, 118, 138, 153, and 180. These PCB congeners were selected as representative indicators for assessing PCB contamination. The mixed standard solution containing these seven indicator PCBs was obtained from Dr. Ehrenstorfer GmbH, a reputable supplier based in Augsburg, Germany.

Thymol, decanoic acid, formic acid, acetic acid, propionic acid, and ChCl were purchased from Energy Chemical Co., Ltd., located in Anqing, China. These chemicals were of reagent grade quality. Chromatographic-grade MeOH, ACN, and acetone were obtained from Sinopharm Chemical Reagent Co., Ltd., based in Shanghai, China. Ultrapure water was prepared by a Milli-Q gradient system manufactured by Millipore, which is located in Bedford, MA, USA.

### 3.2. Instrumentation and Chromatographic Conditions

Chromatographic analysis of the targeted PCBs was conducted by an Agilent 7890A gas chromatograph coupled with an Agilent 5975 mass spectrometer (Agilent, Santa Clara, CA, USA). The system was equipped with an Agilent DB-5ms capillary column (30 m × 0.32 mm × 0.25 μm). Helium gas of high purity was used as the carrier gas at a constant flow rate of 1.2 mL/min. The initial temperature of the column was set at 120 °C and then programmed to increase at a rate of 20 °C/min up to 230 °C. It was then further increased at a rate of 5 °C/min up to 240 °C and held for 3 min. Finally, the temperature was increased at a rate of 5 °C/min up to 260 °C. The total running time for the analysis was 14.5 min. For PCB identification, the samples were analyzed in selective ion monitoring (SIM) mode. The structures and GC-MS conditions for seven indicator PCBs can be found in Table 5. Additionally, Appendix A shows the total ion chromatogram of these PCBs.

An ultrasonic water bath from Ningbo Xinzhi Biotechnology Co., Ltd. in Ningbo, China, model SB-800DT, was utilized for the sonication process. For analysis of the DESs, the Fourier transform infrared (FTIR) spectra were detected on a model Nicolet 6700 spectrometer from Thermo Fisher Scientific (Waltham, MA, USA). Furthermore, the proton nuclear magnetic resonance (^1^H NMR) spectra of the DESs were detected by a Bruker Avance III 600 MHz from Bruker in Fällanden, Switzerland.

### 3.3. Sample Preparation

A stock solution containing a mixture of seven PCBs, each at 10 mg/L, was prepared by dissolving mixed PCB standards in methanol. This stock solution was stored at −20 °C to maintain its stability. Ultrapure water was used to dilute the stock solution to obtain the desired concentration range for spiked samples. During the extraction process, spiked water solutions were prepared immediately before each experiment.

The tap water sample was obtained directly from our laboratory and used without any prior treatment. River water and industrial wastewater samples were collected from a local river and the sewage system of the industrial zone in Taizhou (China), respectively, and they were transported to the laboratory and stored at 4 °C immediately after collection. Before use, each river water sample or industrial wastewater sample was filtered through a 0.45 μm membrane to remove any suspended substances.

### 3.4. DES Preparation

Four hydrophobic DESs (DES1-DES4) were prepared by mixing thymol and decanoic acid at different molar ratios. Specifically, DES1 was prepared by mixing thymol and decanoic acid in a molar ratio of 2:1, DES2 in a ratio of 3:2, DES3 in a ratio of 1:1, and DES4 in a ratio of 1:2. The structures of the hydrophobic DESs are presented in Appendix A.

Additionally, three hydrophilic DESs (DES5-DES7) were prepared using choline chloride (ChCl) and different organic acids. DES5 was prepared by mixing ChCl with formic acid at a molar ratio of 1:2, DES6 with acetic acid at a molar ratio of 1:2, and DES7 with propionic acid at a molar ratio of 1:2. The structures of the hydrophilic DESs have been reported in our previous paper [33].

To prepare these DESs, the respective components were mixed together and stirred using a magnetic stirrer at 80 °C until a transparent clear solution was obtained. Once the mixtures were cooled, they were used in the DLLME procedure for the extraction of PCBs.

### 3.5. Determination of Density and Melting Point

The density of a DES was calculated based on dividing its mass by its volume at room temperature. The method of determining the melting points of DESs was performed using a Mettler Toledo FP62 melting point apparatus at a temperature increase of 0.5 °C/min, as reported elsewhere [29].

### 3.6. DLLME-SHDES-HDESD Procedure

A 12 mL sample solution was transferred into a centrifuge tube containing the mixture of 50 µL of hydrophobic DES2 and 300 µL of hydrophilic DES6. The contents of the centrifuge tube were then subjected to ultrasonic diffusion for 2 min to create a homogeneous emulsion. Following ultrasonic diffusion, the water sample was centrifuged at 4000 rpm for 5 min to separate the aqueous phase and the DES phase. The centrifuge tube was then placed in a refrigerator at a temperature of −20 °C for 5 min to solidify the extracted solvent (upper layer). After solidification, the upper layer was removed from the test tube and allowed to melt at room temperature. Subsequently, a 1 µL aliquot of the melted extracted solvent was collected using a syringe and transferred into the GC-MS instrument for the determination of PCBs. The DLLME-SHDES-HDESD procedure is illustrated in Figure 5.

### 3.7. Enrichment Factor Calculation

The enrichment factor (EF) is calculated from Equation (1):EF = *C_floated_*/*C_o_*(1)
where *C_floated_* and *C_o_* represent the analyte concentration in the extractant and its primary concentration in the sample solution, respectively.

## 4. Conclusions

A novel technique, DLLME-SHDES-HDES, was proposed for the preconcentration of PCBs from water samples. The method utilized the solidification of a hydrophobic DES using a hydrophilic DES as the dispersant. Four hydrophobic DESs (DES1-DES4) and three hydrophilic DESs (DES5-DES7) were tested as potential extractants and dispersants, respectively. The results revealed that DES2, composed of thymol and decanoic acid in a ratio of 3:2, and DES6, composed of ChCl and acetic acid in a ratio of 1:2, exhibited higher extraction efficiency for PCBs. Validation of the developed method demonstrated low LODs, high EFs, and satisfactory reproducibility for PCBs. The developed method offers several advantages, including simplicity, sensitivity, environmental friendliness, and suitability for the determination of residue PCBs in real water samples.

## Figures and Tables

**Figure 1 molecules-29-03480-f001:**
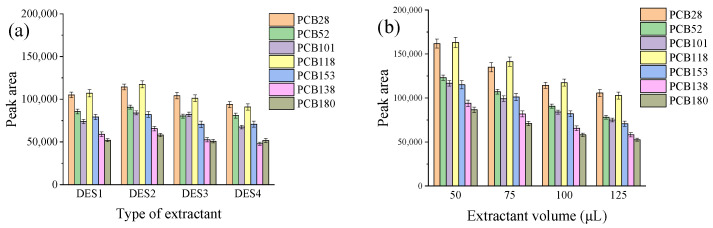
Effect of extractant type (**a**) and extractant volume (**b**) on extraction efficiency. Experimental conditions: 12 mL water sample spiked with 100 ng/L PCBs, 500 µL acetonitrile, ultrasonic diffusion for 3 min; (**a**) extractant volume, 100 µL; (**b**) extractant, 50–125 µL DES2.

**Figure 2 molecules-29-03480-f002:**
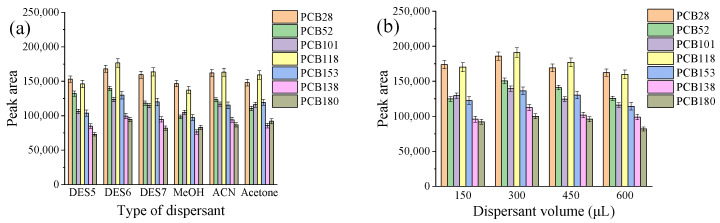
Effect of dispersant type (**a**) and dispersant volume (**b**) on extraction efficiency. Experimental conditions: 12 mL water sample spiked with 100 ng/L PCBs, 50 µL DES2, ultrasonic diffusion for 3 min; (**a**) dispersant volume, 500 µL; (**b**) dispersant, 150–600 µL DES6.

**Figure 3 molecules-29-03480-f003:**
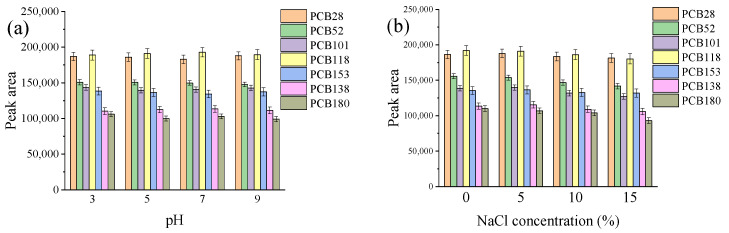
Effect of pH (**a**) and NaCl concentration (**b**) on extraction efficiency. Experimental conditions: 12 mL water sample spiked with PCBs, 50 µL DES2, 300 µL DES6; ultrasonic diffusion for 3 min; (**a**) pH of water sample, 3–9; (**b**) NaCl concentration, 0–15%.

**Figure 4 molecules-29-03480-f004:**
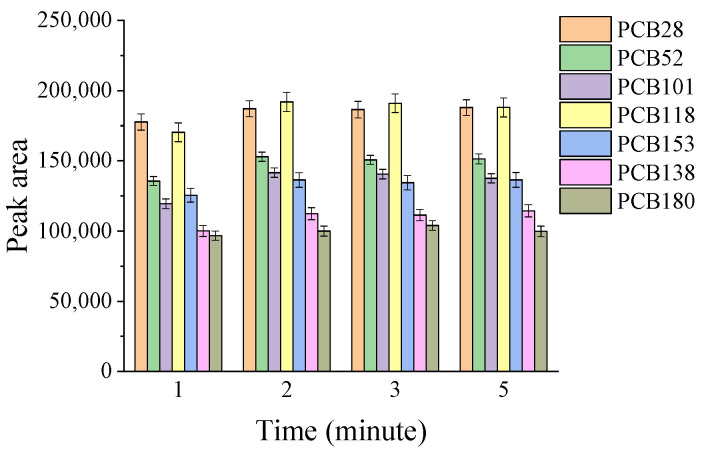
Effect of ultrasonication time on extraction efficiency. Experimental conditions: 12 mL water sample spiked with 100 ng/L PCBs, 50 µL DES2, 300 µL DES6; ultrasonic diffusion for 2–5 min.

**Figure 5 molecules-29-03480-f005:**
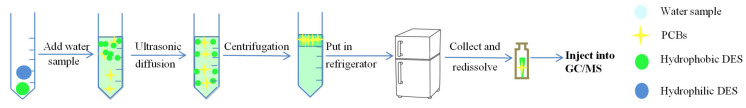
Schematic diagram of the DLLME-SHDES-HDESD Procedure.

**Table 1 molecules-29-03480-t001:** Characterization of the physiochemical properties of the prepared DESs.

DES	[HBA][HBD](Molar Ratio)	*X_HBA_*	*X_HBD_*	Mw ^a^ [g/mol]	ρ [g/cm^3^]	MP ^b^ [°C]
DES1	[thymol][decanoic acid](2:1)	0.67	0.33	157.50	0.9589	19.3
DES2	[thymol][decanoic acid](3:2)	0.60	0.40	159.04	0.9521	18.7
DES3	[thymol][decanoic acid](1:1)	0.50	0.50	161.25	0.9432	17.1
DES4	[thymol][decanoic acid](1:2)	0.33	0.67	164.99	0.9358	18.0
DES5	[ChCl][formic acid](1:2)	0.33	0.67	77.23	1.149	−25.9
DES6	[ChCl][acetic acid](1:2)	0.33	0.67	86.58	1.096	−22.1
DES7	[ChCl][propionic acid](1:2)	0.33	0.67	95.93	1.065	−20.5

Note(s): ^a^ Mw: molar mass, calculated based on Mw = *X_HBA_*·M_HBA_ + *X_HBD_*·M_HBD_, where *X_HBA_*—mole fraction of [HBA], M_HBA_—molar mass of [HBA] [g/mol], *X_HBD_*—mole fraction of [HBD], M_HBD_—molar mass of [HBD] [g/mol]. ^b^ MP—melting point [°C].

**Table 2 molecules-29-03480-t002:** Analytic characteristics of the developed method.

PCB	LR (μg/L)	R^2^	LOD (ng/L)	LOQ (ng/L)	%RSD (*n* = 5)	EF
PCB 28	0.01–5.0	0.9993	3.4	11.4	3.2	193
PCB 52	0.01–5.0	0.9996	3.8	12.7	2.3	199
PCB 101	0.01–5.0	0.9994	4.1	13.8	2.7	182
PCB 118	0.01–5.0	0.9990	3.0	10.1	3.6	187
PCB 138	0.01–5.0	0.9985	4.3	14.3	4.1	185
PCB 153	0.02–5.0	0.9989	4.9	16.2	4.0	204
PCB 180	0.02–5.0	0.9992	5.1	17.0	3.5	195

**Table 3 molecules-29-03480-t003:** Comparison of the developed method with other microextraction methods used for preconcentration and determination of PCBs.

Extraction Method	Detection Method	Matrix	Extractant	Dispersant	LOD (ng/L)	%RSD	EF	Reference
DLLME	GC-MS	Water	Ethyl acetate	Acetone	7.5–15	4.3–7.9	/	[42]
SSME ^a^	GC-MS	Beverages	Switchable forms of heptanoic acid	/ ^e^	2.0–5.0	1.9–4.2	16.2–17.9	[43]
LDS-USAEME ^b^	GC-MS	Water	Isooctane	/	3–12	2.24–3.55	/	[44]
DLLME-SFOD	GC-ECD ^d^	Water	1-undecanol	Acetonitrile	3.3–5.4	5.8–8.8	494–606	[45]
USAEME ^c^	GC-MS	Water	Chloroform	/	14–30	/	/	[46]
DLLME-SHDES- HDESD	GC-MS	Water	DES (thymol/docanoic acid = 3:2, molar ratio)	DES (ChCl/acetic acid = 1:2, molar ratio)	3.0–5.1	2.3–4.1	182–204	This study

Note(s): ^a^ SSME: switchable solvent-based microextraction. ^b^ LDS-USAEME: low density solvent-ultrasound-assisted emulsification microextraction. ^c^ USAEME: ultrasound-assisted emulsification-microextraction. ^d^ GC-ECD: gas chromatography–electron capture detector. ^e^: not referred.

**Table 4 molecules-29-03480-t004:** Analysis of real water samples under optimal conditions.

PCB		Tap Water		River Water		Industrial Water	
	Spiked (μg/L)	Found (μg/L)	Spiked Recovery (%, *n* = 3)	Found (μg/L)	Spiked Recovery (%, *n* = 3)	Found (μg/L)	Spiked Recovery (%, *n* = 3)
PCB 28	0	nd ^a^	nd	nd	nd	nd	nd
	0.1	0.0996	99.60 ^b^ ± 2.9 ^c^	0.1028	102.80 ± 3.1	0.0985	98.50 ± 3.2
	1	1.0132	101.32 ± 3.4	0.9718	97.18 ± 3.2	0.9679	96.79 ± 2.7
PCB 52	0	nd	nd	nd	nd	0.36	nd
	0.1	0.0936	93.60 ± 2.1	0.0963	96.30 ± 2.6	0.4586	98.60 ± 2.5
	1	0.9584	95.84 ± 2.2	0.9258	92.58 ± 2.5	1.2939	93.39 ± 2.7
PCB 101	0	nd	nd	nd	nd	nd	nd
	0.1	0.1022	102.20 ± 2.8	0.0964	96.40 ± 3.2	0.1015	101.50 ± 3.1
	1	0.9588	95.88 ± 3.0	1.0302	103.02 ± 2.7	0.9963	99.63 ± 3.3
PCB 118	0	nd	nd	nd	nd	nd	nd
	0.1	0.1014	101.40 ± 3.8	0.1039	103.90 ± 4.2	0.0992	99.2 ± 3.7
	1	1.0022	100.22 ± 3.5	0.9712	97.12 ± 3.9	1.0187	101.87 ± 4.0
PCB 138	0	nd	nd	nd	nd	0.17	nd
	0.1	0.0962	96.20 ± 4.3	0.0934	93.40 ± 3.7	0.2659	95.90 ± 3.8
	1	0.9429	94.29 ± 3.8	0.9853	98.53 ± 4.2	1.1432	97.32 ± 4.5
PCB 153	0	nd	nd	nd	nd	nd	nd
	0.1	0.1028	102.80 ± 3.5	0.0986	98.60 ± 4.3	0.0977	97.70 ± 4.1
	1	0.9636	96.36 ± 4.1	1.0070	100.70 ± 3.9	1.0138	101.38 ± 3.8
PCB 180	0	nd	nd	nd	nd	nd	nd
	0.1	0.0996	99.60 ± 3.5	0.1012	101.20 ± 3.1	0.0981	98.1 ± 3.7
	1	0.9532	95.32 ± 3.1	1.0034	100.34 ± 3.6	0.9682	96.82 ± 3.2

Note(s): ^a^ Not detected (<LODs). ^b^ Mean of three determinations. ^c^ Standard deviation for three determinations.

**Table 5 molecules-29-03480-t005:** The structures and GC-MS conditions for the targeted PCBs.

PCB	Structure	Retention Time (min)	Selected Ions (*m*/*z*)
PCB 28	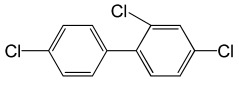	6.382	255.8 *, 185.9, 149.9, 257.8
PCB 52	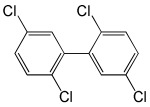	6.795	291.8 *, 219.8, 254.8, 183.9
PCB 101	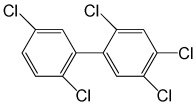	8.072	325.8 *, 253.9, 183.8, 290.8
PCB 118	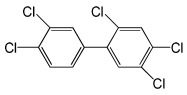	9.346	325.8 *, 253.8, 183.9, 323.8
PCB 153	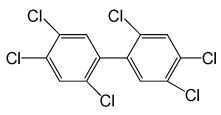	9.863	359.8 *, 289.8, 324.9, 361.8
PCB 138	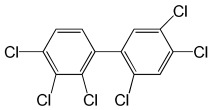	10.632	359.7 *, 289.8, 324.7, 144.9
PCB 180	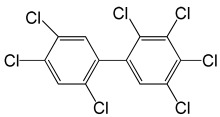	12.725	393.7 *, 323.7, 253.9, 161.7

Note(s): * Quantitative ion.

## Data Availability

The raw data supporting the conclusions of this article will be made available by the authors on request.

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
