# Peer review of "A Novel DLLME Method Involving a Solidifying Hydrophobic Deep Eutectic Solvent Using Hydrophilic Deep Eutectic Solvent as the Dispersant for the Determination of Polychlorinated Biphenyls in Water Samples"

_molecules, 2024, doi:10.3390/molecules29153480_

Round 1
Reviewer 1 Report
Comments and Suggestions for Authors
Manuscript “A novel dispersive liquid-liquid microextraction method based on solidification of hydrophobic deep eutectic solvent using hydrophilic deep eutectic solvent as dispersant for the determination of polychlorinated biphenyls in water samples”.
The authors used different DES as extractants or dispersants in order to quantify different PCBs, which is an interesting topic given the global concern about their concentration problems in the food chain. Although interesting results are shown, there are still several doubts about the real effect of DES on the concentration of the analyte. I recommend this publication for major revisions to see if the authors can address all the comments.
1. Line 67. There are many papers claiming that some DES are more toxic than some common ILs. I recommend rewriting this so as not to raise false expectations about these solvents.
2. The DES abbreviation must be indicated in the text, table, figure and not with DES1, DES2, that is more complex to find out a correlation between the DES and the PCB recovery.
3. In the section "2.1. Characterization of DESs" I do not understand why it performs the characterization of the thymol-decanoic acid mixture since it is an eutectic mixture quite studied in literature, so the NMR and FTIR study does not provide more relevant information. This should be supplementary material. On the other hand, if it would be interesting and worth discussing in the analysis, is that how do you know if the 3:2 ratio you tested really all the molecules participate in the DES and there is not some molecule solvated by the DES, i.e., that an excess is solubilized in the DES. The donor and acceptor groups of the molecules and according to the ratio chosen do not fully bind all the polar zones (check sigma profile).
4. Table 1. How was the melting point obtained? If it was from the literature, it must be cited, on the other hand if you performed the experiment, it must be written in the procedure.
5. Line 143. It is incorrect to say that the lower the volume the lower the extraction, although it will be more concentrated and that is what is shown in 2B, this does not mean that the greatest amount was extracted, since this also depends on the volume used.
6. Line 157, how does the DES part promote extraction? They should come up with a mechanism to help understand the effect of the dispersant. Why not just add that component and not the whole DES to save operational costs and make the process greener? You should show FT-IR or NMR assays after extraction to know if the DES6 persists as such or is completely separated, otherwise it does not make much sense to add the DES as such.
7. Did you perform an FTIR or NMR test on the recovered solidified sample to see if it contained dispersant?
8. Figure 2 and Figure 3 should be in concentration (or extraction) and not in area. In this way it is easier to compare with other works.
9. Line 185, in %w/w or %mol/mol? specify the unit of concentration.
10. Line 189. But viscosity should not be a constraint, since their method should be at equilibrium. If it is not all your previous results are affected by mass transfer and thus are not valid.
11. “2.4. Effect of Ultrasound Time” this should be the first thing to be studied to have valid results.
12. Line 234. You should mention what specifically you are basing this on, have LD50 studies been done? biodegradability? LCA? etc.
13. Table 4. I am surprised that all the samples have almost the same concentration no matter the source, is this correct? taking into account the sample from the industrial sector should have more PCBs? or there are no industries that work with Chlorine and at high temperatures?
14. Based on the above. Did you test blanks to study if there is any trace of PCBs in your DES or other reagents? It must be included in the supplementary data.
15. I also wonder if they tested only the dispersant to know its concentration factor by itself. This is important since it does not show the mechanism of PCB extraction.
16. The purity of the reagents or solvents used must be indicated in the experimental section.
Author Response
Comments 1: Line 67. There are many papers claiming that some DESs are more toxic than some common ILs. I recommend rewriting this so as not to raise false expectations about these solvents.
Response 1: Thanks! We have rewritten the content of line 67.
Comments 2: The DES abbreviation must be indicated in the text, table, figure and not with DES1, DES2, that is more complex to find out a correlation between the DES and the PCB recovery.
Response 2: Thanks! The abbreviation "DES" has been defined in the abstract and introduction. For ease of description, the four hydrophobic DESs are referred to as DES1-DES4, and the hydrophilic DESs are referred to as DES5-DES7. The components of these DESs are listed in Table 1. Additionally, the extraction efficiency is evaluated by comparing the peak areas of PCBs, rather than PCB recovery.
Comments 3: In the section "2.1. Characterization of DESs" I do not understand why it performs the characterization of the thymol-decanoic acid mixture since it is an eutectic mixture quite studied in literature, so the NMR and FTIR study does not provide more relevant information. This should be supplementary material. On the other hand, if it would be interesting and worth discussing in the analysis, is that how do you know if the 3:2 ratio you tested really all the molecules participate in the DES and there is not some molecule solvated by the DES, i.e., that an excess is solubilized in the DES. The donor and acceptor groups of the molecules and according to the ratio chosen do not fully bind all the polar zones (check sigma profile).
Response 3: Thanks! The position of Figure 1 (FTIR and 1H NMR) has been moved to the supplementary materials.
All polar protic solvents have strong hydrogen bonds. A DES is a eutectic mixture that shows a negative melting point deviation from the ideal eutectic (10.1007/s10953-018-0793-1). This deviation can also be caused by the formation of hydrogen bonds. This is why DES is such at all the ratios used, including the 3:2 ratio.
Comments 4: Table 1. How was the melting point obtained? If it was from the literature, it must be cited, on the other hand if you performed the experiment, it must be written in the procedure.
Response 4: Thanks! The method for determining the melting point has been detailed in section 3.5 (Determination of Density and Melting Point).
Comments 5: Line 143. It is incorrect to say that the lower the volume the lower the extraction, although it will be more concentrated and that is what is shown in 2B, this does not mean that the greatest amount was extracted, since this also depends on the volume used.
Response 5: Thanks! This statement has been corrected.
Comments 6: Line 157, how does the DES part promote extraction? They should come up with a mechanism to help understand the effect of the dispersant. Why not just add that component and not the whole DES to save operational costs and make the process greener? You should show FT-IR or NMR assays after extraction to know if the DES6 persists as such or is completely separated, otherwise it does not make much sense to add the DES as such.
Response 6: Thanks! In the first paragraph of section 2.2.2 (Effect of Dispersant Type and Dispersant Volume), we have described the role of the hydrophilic DES (dispersant) in the extraction process as follows: During the experimental process, the hydrophilic DES was observed to decompose upon the addition of the water sample. This decomposition facilitates the dispersion of the extractant, leading to enhanced extraction efficiency. To aid in understanding the PCB extraction mechanism, the DLLME-SHDES-HDESD procedure is illustrated in Figure 5.
The dispersant DES6 is synthesized from choline chloride and acetic acid. Upon adding it to the water sample, DES6 enters the aqueous solution and causes the extractant to disperse and form droplets, similar to the principle of dispersants in many other microextraction methods. Moreover, DES6 decomposes after entering the aqueous solution, enhancing its dispersion effect on the extractant. Therefore, DES6 can achieve better extraction results when used as a dispersant. We believe that the primary reason DES6 can promote the formation of micro-droplets in the extractant is due to its ability to dissolve with both the extractant and water, rather than its decomposition in the aqueous solution. In microextraction, dispersants can dissolve with extraction solvents and aqueous solutions. When using organic dispersants such as methanol, acetonitrile, and acetone, these dispersants do not decompose during the extraction process. Moreover, after the extraction process is completed, these organic dispersants will remain in a certain proportion in both the aqueous solution and the extraction liquid, but this does not significantly impact the extraction effect. Therefore, we believe that whether DES6 is completely decomposed during the extraction process and whether a small amount of DES6 is present in the extraction solution will not significantly impact the extraction efficiency.
Comments 7: Did you perform an FTIR or NMR test on the recovered solidified sample to see if it contained dispersant?
Response 7: Thanks! In this study and our previous studies, we performed FTIR and NMR tests on the recovered solidified samples to check for the presence of the dispersant. The results showed that no hydrophilic DES or its components were detected in the extracted solvent. However, according to response of comments 6, it will not significantly impact the extraction efficiency even if a small amount of DES6 is present in the recovered solidified sample.
Comments 8: Figure 2 and Figure 3 should be in concentration (or extraction) and not in area. In this way it is easier to compare with other works.
Response 8: Thanks! In DLLME, the extraction efficiency is generally evaluated in terms of the peak area or recovery of the analytes, and different studies are often compared in terms of their detection limits and repeatability. (References 21, 36-46)
Comments 9: Line 185, in %w/w or %mol/mol? specify the unit of concentration.
Response 9: Thanks! The percentage on line 185 is %w/v; we have added the unit to the concentration.
Comments 10: Line 189. But viscosity should not be a constraint, since their method should be at equilibrium. If it is not all your previous results are affected by mass transfer and thus are not valid.
Response 10: Thanks! We have corrected this statement. With the help of another reviewer, we speculated that this phenomenon was caused by the salting-out effect.
Comments 11: “2.4. Effect of Ultrasound Time” this should be the first thing to be studied to have valid results.
Response 11: Thanks! The type and volume of the extractant, as well as the type and volume of the dispersant, are crucial factors that can affect extraction efficiency. The effect of ultrasound time is also influenced by these factors. Therefore, these factors are the primary focus in nearly all DLLME methods.
Comments 12: Line 234. You should mention what specifically you are basing this on, have LD50 studies been done? biodegradability? LCA? etc.
Response 12: Thanks! The thymol-decanoic acid eutectic mixture is quite studied in literatures. According to these literatures, the thymol-decanoic acid eutectic mixture is considered green solvents. Meanwhile, the dispersant DES6 will decompose during the extraction process, producing choline chloride and acetic acid, both of which are low-toxicity substances. Therefore, we believe that the extractants and dispersants used in our developed method are environmentally friendly.
Comments 13: Table 4. I am surprised that all the samples have almost the same concentration no matter the source, is this correct? taking into account the sample from the industrial sector should have more PCBs? or there are no industries that work with Chlorine and at high temperatures?
Response 13: Thanks! The experimental results are accurate, and we conducted three replicates for each sample. The results also indicate that the matrix effect is minimal when measuring polychlorinated biphenyls (PCBs) in different water samples. Only two types of PCBs, PCB52 and PCB138, were detected in industrial wastewater samples, and their concentrations were relatively low. This phenomenon may be related to the fact that the factories are mostly biological enterprises.
Comments 14: Based on the above. Did you test blanks to study if there is any trace of PCBs in your DES or other reagents? It must be included in the supplementary data.
Response 14: Thanks! In our previous test blanks, no trace of PCBs was detected in the prepared DESs and their components. In the second paragraph of section 3.1 (Materials and Reagents), we emphasized that all chemicals used for DES preparation are of reagent grade quality, so the results of the test blanks are theoretically reasonable. These data have not been reported in almost all publications on DLLME approaches using DES as extractant or dispersant. Therefore, this result was not included in the data of our paper.
Comments 15: I also wonder if they tested only the dispersant to know its concentration factor by itself. This is important since it does not show the mechanism of PCB extraction.
Response 15: Thanks! In our developed method, the target PCBs are extracted into the extractant (hydrophobic DES). The role of the hydrophilic DES (dispersant) in the extraction process was described in response to question 6. To aid in understanding the mechanism of PCB extraction, the mechanism has been illustrated in Figure 5.
Comments 16: The purity of the reagents or solvents used must be indicated in the experimental section.
Response 16: Thanks! The purity of the reagents or solvents has been indicated in the second paragraph of section 3.1 (Materials and Reagents).

Reviewer 2 Report
Comments and Suggestions for Authors
Author Response
Comments 1: The similarity with the literature is too high, the iThenticate report from molecules identified 35% similarity, this number needs to be reduced too much.
Response 1: Thanks! We have rewritten some parts of the article to reduce similarity with existing literature, and the rewritten parts are marked in red.
Comments 2: The title is too long, do the authors think they can't reduce it?
Response 2: Thanks! The title of the article has been slightly modified.
Comments 3: Please be clear in the abstract which type of HDES and DES was used.
Response 3: Thanks! Lines 19-22 in the abstract describe a hydrophobic DES called DES2 and a hydrophilic DES named DES6.
Comments 4: Hydrophobic DES have been widely reported in the literature, so the authors have sufficient basis through these works to be able to improve the literature review of hydrophobic as well as hydrophilic DES.
Response 4: Thanks! The hydrophobic DESs have been reviewed more extensively in the fifth paragraph of the introduction.
Comments 5: Topic 2.1 mentions DES5, 6 and 7. However, I don't see which DES these are. The table below only lists 1 to 4.
Response 5: Thanks! The hydrophilic DESs (DES5, DES6, and DES7) have been added to Table 1. In the previously submitted article (lines 308-311), the components of these hydrophilic DESs were only described in section 3.4 (DESs Preparation).
Comments 6: Instead of Table 1, I suggest creating a solid-liquid diagram, which is more suitable for better visualization.
Response 6: Thanks! A solid-liquid phase diagram for a mixture of thymol and decanoic acid has been created (Figure S3).
Comments 7: A discussion with the literature needs to be held. I don't see that in the results. I couldn't see any work being discussed in this topic. This is corroborated by the manuscript's low number of citations.
Response 7: Thanks! A supplementary discussion on the experimental results has been conducted, and additional supporting articles have been provided as references.

Reviewer 3 Report
Comments and Suggestions for Authors
The intotlated work "A novel dispersive liquid-liquid microextraction method based on solidification of hydrophobic deep eutectic solvent using hydrophilic deep eutectic solvent as dispersant for the determination of polychlorinated biphenyls in water samples" describes the use of DES for the development of an analytical method for the determination of PCBs. The subject is interesting, and the replacement of VOCs with sustainable solvents is also of interest for analytical purposes. Of interest is the validation of the method and comparison with other methods reported in the literature. However, the work has some critical issues that need to be resolved before publication.
First of all, in the introduction, the definition of DES is wrong. The formation of hydrogen bonds does not identify DES as such. All polar protic solvents have strong hydrogen bonds. A DES is a eutectic mixture that shows a negative melting point deviation from the ideal eutectic (10.1007/s10953-018-0793-1). This deviation can also be caused by the formation of hydrogen bonds. This is why DES is such at all the ratios used. The molar ratio must be specified for each DES. Based on the definition, it is conceptually wrong to call DES 1-4 what are in fact the usual DES. The authors must amend this part. In addition, DESs are used in many processes for the extraction of polyphenols (10.1039/D4GC00526K), metals (10.1039/D1GC03851F), ligin (10.3389/fchem.2023.1270221), pollutants (10.1016/j.jece.2022.108574). Some examples of these applications must be given.
In addition, the structures of the analysed PCBs must be reported. The same applies to the structures of the DES used. For DES 5-7 it is necessary to go into the experimental part to know which DES are.
In addition, an extraction scheme would help in understanding the analytical procedure. In fact, it is not clear what the "Dispersant" is. Is it a cosolvent, in which case a ternary DES is being formed with a different structure from the original. Are hydrophilic DESs miscible with hydrophobic ones? In addition, there is a transfer of the hydrophilic DES to the aqueous phase. This will contribute to the salting out reported in section 2.2.3. I beg the authors to clarify this because the role of the various components is not clear.
Another problem is the use of basic pH. The DES impeigated are all acidic and the basic pH salts the acid decomposing the DES. This has an effect on the eventual recycling in case it is necessary to do so. The authors comment on this aspect.
Only after clarifying these aspects will the work be suitable for publication.
Comments on the Quality of English Language
Moderate editing of English language required
Author Response
Comments 1: In the introduction, the definition of DES is wrong. The formation of hydrogen bonds does not identify DES as such.
Response 1: Thanks! The definition of DES has been revised in the introduction.
Comments 2: It is conceptually wrong to call DES 1-4 what are in fact the usual DES.
Response 2: Thanks! For convenience of description, four hydrophobic DESs are named as DES1-DES4, and their components have been listed in Table 1. This approach has also been reported in other literature (https://doi.org/10.1002/jssc.202100590).
Comments 3: DESs are used in many processes for the extraction of polyphenols (10.1039/D4GC00526K), metals (10.1039/D1GC03851F), ligin (10.3389/fchem.2023.1270221), pollutants (10.1016/j.jece.2022.108574).
Response 3: Thanks! The hydrophobic DESs have been reviewed more extensively in the fifth paragraph of the introduction, with some examples of their applications provided.
Comments 4: The structures of the analysed PCBs must be reported. The same applies to the structures of the DES used.
Response 4: Thanks! The structures of the analyzed PCBs have been added to Table 5. The structures of the DESs used are described in section 3.4 (DESs Preparation). The structures of hydrophobic DESs are presented in Figure S4. The structures of hydrophilic DESs have been reported in our previous paper (Reference 33, https://doi.org/10.3390/w15142579).
Comments 5: For DES 5-7 it is necessary to go into the experimental part to know which DES are.
Response 5: Thanks! The hydrophilic DESs (DES5, DES6, and DES7) have been added to Table 1. In the previously submitted article, the components of the hydrophilic DESs were only described in section 3.4 (DESs Preparation) (lines 308-311).
Comments 6: An extraction scheme would help in understanding the analytical procedure. In fact, it is not clear what the "Dispersant" is.
Response 6: Thanks! We have created a schematic diagram (Figure 5) to describe the microextraction procedure.
Comments 7: Are hydrophilic DESs miscible with hydrophobic ones?
Response 7: Thanks! At room temperature, any of hydrophilic DESs (DES1, DES2, DES3 and DES4) can miscible with any of hydrophobic ones (DES5, DES6 and DES7).
Comments 8: There is a transfer of the hydrophilic DES to the aqueous phase. This will contribute to the salting out reported in section 2.2.3.
Response 8: Thanks! We have revised the statement in section 2.2.3, and speculated that this phenomenon was caused the salting-out effect.
Comments 9: Another problem is the use of basic pH. The DES implicated are all acidic and the basic pH salts the acid decomposing the DES. This has an effect on the eventual recycling in case it is necessary to do so.
Response 9: Thanks! Under experimental conditions, the extraction time was within 5 minutes (excluding centrifugation time), and no decomposition of DES was detected under weakly alkaline conditions. This may be due to two reasons. Firstly, DES may have some resistance to weakly alkaline environments. Secondly, the degradation of hydrophilic DES in water may produce acidic byproducts, which can lower the pH value of the water sample, potentially providing a protective effect on the DES.

Round 2
Reviewer 1 Report
Comments and Suggestions for Authors
The authors have amended the observations and this work can be published
Author Response
Comment 1: The authors have amended the observations and this work can be published.
Response1: Thanks!
Reviewer 3 Report
Comments and Suggestions for Authors
I would like to thank the authors for accepting my suggestions. However, the answer on the pH part is not satisfactory. The addition of a base to the acid DES leads to the neutralisation of the acid itself and thus changes the structure of DES. This reaction is not reversible and therefore DES is not stable in the indicated range 6-8. This must be pointed out. Since these are micro-extractions, recycling of the DES is not a priority, but for all intents and purposes it is not the indicated DES that is being extracted but the one that contains the salt of the acid initially present.
Author Response
Comment 1: However, the answer on the pH part is not satisfactory. The addition of a base to the acid DES leads to the neutralisation of the acid itself and thus changes the structure of DES. This reaction is not reversible and therefore DES is not stable in the indicated range 6-8. This must be pointed out. Since these are micro-extractions, recycling of the DES is not a priority, but for all intents and purposes it is not the indicated DES that is being extracted but the one that contains the salt of the acid initially present.
Response 1: Thanks! Previous studies have investigated the effect of pH on extraction efficiency when deep eutectic solvents (DESs) are used as extractants. For instance:
Literature 1: The pH of the spiked solution ranged from 1 to 8, and the DES was prepared by combining L-menthol and lactic acid in a molar ratio of 1:2. (https://doi.org/10.1002/jssc.202200110).
Literature 2: The pH of the spiked solution ranged from 5 to 9, and the DES was prepared by combining thymol and octanoic acid in a molar ratio of 1:4. (https://doi.org/10.1039/d0ay02121k, Reference 37).
Literature 3: The pH of the spiked solution ranged from 3 to 11, and the DES was prepared by combining thymol and octanoic acid in a molar ratio of 1:5. (https://doi.org/10.1016/j.foodchem.2020.126424).
The experimental data from these studies indicated that pH value had almost no significant effect on the extraction efficiency. In our study, we observed a similar result when the pH of the water samples ranged from 6 to 8.
The pH values of tap water and river water generally fall between 6.5 and 8.5. Previous studies using DESs as extractants have shown that there is no need to adjust the pH of these real water samples before microextraction. Additionally, no reports of DES degradation during the microextraction process were found in these studies. For instance:
Literature 1: Real samples (tap water and industrial wastewater) were studied, and the DES was prepared by combining choline chloride and phenol in a molar ratio of 1:2. (https://doi.org/10.1016/j.chroma.2015.11.007)
Literature 2: Real samples (river water, lake water and well water) were studied, and the DES was prepared by combining tetra-n-butylammonium bromide (TBAB) and decanoic acid in a molar ratio of 1:2. (https://doi.org/10.1007/s10337-018-3548-7)
Literature 3: Real samples (tap water and well water) were studied, and the DES was prepared by combining choline chloride and oxalic acid in a molar ratio of 1:2. (https://doi.org/10.1016/j.chroma.2020.461618)
Based on these findings, we speculate that hydrophobic DESs may have some resistance to weakly acidic or weakly alkaline environments, and thus, the pH of the water samples has no significant effect on the extraction efficiency in our study. Furthermore, we observed that the degradation of a hydrophilic DES (DES6) in water may produce acidic byproducts. These byproducts could lower the pH value of the water sample, potentially providing a protective effect on the DES.